# Study of Protein-Protein Interactions in Septin Assembly: Multiple amphipathic helix domains cooperate in binding to the lipid membrane

S. Mahsa Mofidi[1,2,3], Abhilash Sahoo[1,2], Christopher J. Edelmaier[1,3],
Stephen J. Klawa[3], Ronit Freeman[3], Amy Gladfelter[4], M. Gregory Forest[3,5],
Ehssan Nazockdast[3], Sonya M. Hanson[1,2*]

1 Center for Computational Biology, Flatiron Institute, New York City, New York, United States of America,
2 Center for Computational Mathematics, Flatiron Institute, New York City, New York, United States of America, 3 Department of Applied Physical Sciences, The University of North Carolina at Chapel Hill, Chapel Hill, North Carolina, United States of America, 4 Department of Cell Biology, Duke University, Durham, North Carolina, United States of America, 5 Department of Mathematics, The University of North Carolina at Chapel Hill, Chapel Hill, North Carolina, United States of America

* shanson@flatironinstitute.org

## Abstract

Septins are a conserved family of cytoskeletal proteins known for sensing micron-scale membrane curvature via amphipathic helix (AH) domains. While cooperative interactions in septin assembly have been suggested, the molecular mechanisms governing membrane binding and assembly remain unclear. Building on prior findings, we use all-atom molecular dynamics simulations to examine how single and paired extended AH domains, derived from Cdc12, interact with lipid bilayers. We find that a single membrane-bound AH adopts a bent conformation upon membrane association. In solution, a second AH peptide preferentially interacts with the bound peptide through conserved salt bridges, favoring an antiparallel arrangement. Simulations of covalently linked AH tandems confirm the stability of this configuration. When two AH domains are membrane-bound, they induce localized lipid packing defects, reduce tail order, and exhibit slight peptide displacement on planar bilayers. These observations suggest a cooperative AH binding mechanism and are consistent with models in which lipid packing defects facilitate multivalent AH engagement in curved membrane environments. Our findings advance the mechanistic understanding of septin-membrane interactions and highlight the role of cooperative AH domain binding in stabilizing higher-order structures.

## Author summary

Cells sense and respond to their shape in order to grow, divide, and organize internal structures. Septins are a family of proteins that help cells detect membrane curvature, allowing them to assemble at specific cellular locations. A key

**Data availability statement:** Information to find all data and code supporting the findings of this study are available in the main text or the supplementary materials. Custom analysis code is publicly available at https://github.com/flatironinstitute/septin_AH_analysis. Full molecular dynamics simulation trajectories and associated input files have been deposited in an open-access Zenodo repository at https://doi.org/10.5281/zenodo.16790335.

**Funding:** We acknowledge support from the Alfred P. Sloan Foundation Matter-to-Life (grant no. G-2021-14197). R.F. acknowledges financial support in the form of a Cottrell Scholar Award (CS-CSA-2023-033), sponsored by Research Corporation for Science Advancement. The funders had no role in study design, data collection and analysis, decision to publish, or preparation of the manuscript.

**Competing interests:** The authors have declared that no competing interests exist.

component of this process is the amphipathic helix, a short protein segment that is inserted into membranes. While previous studies suggested that these helices may work together, the details of this cooperation were not well understood. Using computer simulations, we explore how helices from yeast septins interact with each other and with lipid membranes. We found that a single helix bends into a curved shape upon binding to the membrane and that a second helix preferentially associates with it in a specific arrangement. When two helices are present, they locally alter membrane structure in ways that can favor curved membrane environments. These results show how small, cooperative interactions can help septins sense and stabilize membrane shape, providing insight into how cells organize their internal architecture.

## Introduction

Septins are a conserved family of GTP-binding proteins that play essential roles in various cellular processes, including cell division, membrane remodeling, and cytoskeletal organization [1,2]. A hallmark of septin function is their ability to sense and respond to membrane curvature, a property critical for maintaining cellular architecture and enabling dynamic shape changes [3]. Septins polymerize into filaments and higher-order assemblies [4] that can scaffold and remodel cellular membranes [5]. In *Saccharomyces cerevisiae*, septins form hetero-oligomeric complexes that polymerize along the membrane [6,7]. Previous studies revealed that amphipathic helices (AHs) at the C-terminal end of the Cdc12 subunit shallowly insert into the lipid bilayer and facilitate membrane binding [8]. Experimental studies have shown that septins' AH domains bind to membranes and sense micron-scale curvature [9,10]. A kinetic model showed septins undergo a multi-step assembly process, and curvature preference is modulated by protein concentration and membrane geometry [11]. Although kinetic models suggest cooperative septin-membrane interactions, the molecular mechanisms underlying this cooperativity remain unresolved. Moreover, septins may sense membrane curvature and deform membranes, facilitating additional protein binding—a feedback process observed in synthetic systems [12,13]. This membrane-deforming ability is likely due to the multivalent arrangement of AH domains, which serve as tether points for septin scaffold assembly. Interestingly, while native septins display curvature preference, an isolated AH domain lacks curvature sensitivity [14]. However, tandem AH constructs can restore the function [15], pointing to the importance of multivalent interactions and flanking effects in membrane binding and peptide configuration.

Amphipathic helices are short α-helical motifs characterized by a spatial separation of hydrophobic and hydrophilic residues, enabling them to insert into lipid bilayers and act as sensors or inducers of curvature [16,17]. This structural feature underlies their widespread role in membrane dynamics across many protein families. For example, in BAR proteins, amphipathic helices contribute to the organization of higher-order structures [18]. Association with the membrane stabilizes the AH helical

structure, primarily through interactions between hydrophobic residues and lipid tails [19]. AH–membrane interactions are also influenced by membrane curvature, lipid composition, and the presence of defects—regions where lipid tail exposure facilitates peptide insertion. Membrane defects on convex surfaces enhance AH folding, promoting its helical structure and altering defect distribution [20,21]. Membrane defects, defined as exposed hydrophobic lipid tails, play a crucial role in protein engagement. Additionally, studies indicate that lipid composition and membrane curvature influence defect formation, thereby regulating protein recruitment [22,23]. Although AH domains in various proteins are known to facilitate membrane binding, the specific mechanisms of binding and the contribution of multiple AH domains to the higher-order assembly remain poorly understood. Building upon the findings of our previous study [14], which showed that the helicity and charge distribution of isolated septin AH domains tune their membrane-binding affinity, this study explores how multiple extended AH domains interact in more complex, multivalent membrane-associated contexts. While our previous work focused on isolated amphipathic helices and their individual membrane-binding properties, the present study extends this framework in several important ways. First, we investigate extended AH domains that include conserved N- and C-terminal flanking regions, revealing bending behavior and inter-peptide interaction motifs that are absent in shorter constructs. Second, we explicitly examine cooperative effects by simulating systems containing multiple AH domains, both as independent peptides and as covalently linked tandem constructs. This allows us to directly probe peptide–peptide interactions and their role in stabilizing membrane association. Third, we characterize how multiple AH domains jointly reorganize the surrounding lipid environment, inducing localized packing defects, changes in acyl-chain order, and modest peptide displacement that are not accessible in single-peptide simulations. Together, these advances move beyond single-helix analyses and provide new mechanistic insight into cooperative AH-mediated membrane interactions relevant to septin assembly.

To address the unresolved question of how AH domains contribute to cooperative membrane binding and septin assembly, we used molecular dynamics (MD) simulations to investigate the behavior of the extended AH domain from Cdc12 from the *S. cerevisiae* septin complex. We included upstream and downstream residues to more closely reflect the native context of the Cdc12 C-terminal region (Fig 1A). Simulations were initiated from two configurations: with the peptide placed in the solution above the membrane (unbound; Fig 1B) or shallowly inserted into the bilayer (bound; Fig 1C). Throughout the manuscript, peptides initialized as in panel (B) are referred to as 'floating' peptides, while those initialized as in panel (C) are referred to as 'bound' peptides. We examined both single-peptide and two-peptide systems to explore potential cooperative effects. We begin by characterizing the structural behavior of a single AH peptide that interacts with planar lipid bilayers. Then, we analyzed multi-peptide simulations to assess how interpeptide interactions may influence membrane association. The results suggest that the presence of a membrane-bound peptide can modulate the orientation and positioning of another peptide, likely through specific stabilizing interactions. Finally, we simulated covalently linked AH domains—referred to as tandem constructs [15] —to examine how connectivity influences structural stability. While palindromic CN-NC configurations exhibit instability, containing charged C-terminal domains stabilized the extended tandem structure. These extended tandem AHs may represent a simple model of septin with a minimal linker between two AH domains and the C-termini of Cdc12. Together, these simulations provide molecular-level insights into how AH domains may cooperatively engage membranes and offer a framework for understanding or engineering membrane-binding peptides with tunable properties.

## Results

### The extended AH peptide adopts a stable, curved conformation when bound to the membrane.

We began by analyzing the behavior of a single membrane-bound extended AH peptide composed of 34 residues, including N- and C-terminal extensions flanking the core amphipathic helix (AH). Across all four simulation replicas, the bound peptide consistently adopted a stable, curved, smile-like conformation throughout the trajectory, as shown in a simulation snapshot in Fig 2A. This curvature appears to be a consequence of the peptide's domain-specific interaction

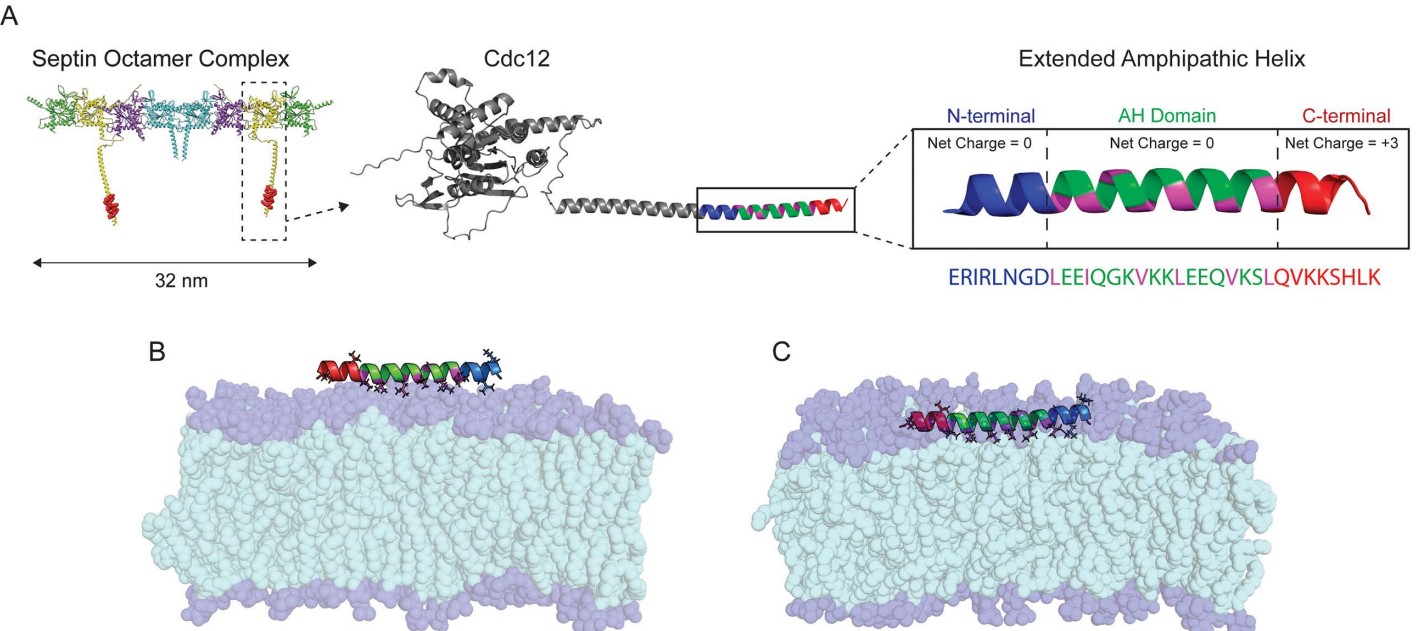

**Fig 1. Septin structure and initial configurations of the extended AH with a lipid bilayer. (A)** PDB structure of a nanoscale septin oligomer with a zoomed-in view of the Cdc12 subunit and its Amphipathic Helix (AH) domain with adjacent residues. The extended AH regions are color-coded: N-terminus (blue), AH domain (green), hydrophobic residues (magenta), and C-terminus (red). The corresponding amino acid sequence is also shown. **(B)** Initial configuration of the unbound state: the extended AH peptide is placed in solution above the lipid bilayer. **(C)** Initial configuration of the bound state: the extended AH is shallowly inserted into the membrane. The AH hydrophobic face is aligned with the lipid tails. Lipid hydrophobic tails are shown in cyan and phosphate headgroups in dark blue. Lipids are rendered semi-transparent to highlight the peptide orientation and residue types.

with the membrane. The central ~18 residues correspond to the core amphipathic helix (AH), which exhibits a clear spatial segregation of hydrophobic and polar side chains, enabling it to stably associate with the membrane surface via hydrophobic insertion. In contrast, the N- and C-terminal extensions flanking the AH do not follow a regular amphipathic pattern and lack the structural features required for stable membrane insertion. As a result, the central AH region remains closely bound to the membrane surface, while the terminal segments are lack this pattern and tend to bend upward toward the aqueous phase. This asymmetric interaction along the peptide length generates a net bending moment, resulting in the observed smile-like conformation. Supporting this interpretation, our previous simulations of the isolated AH segment alone (without the flanking regions) do not exhibit significant curvature [14], highlighting the importance of the full peptide architecture in driving this membrane-bound conformation. The presence of the bound peptide perturbs the surrounding lipid environment. The area per lipid (APL) map (Fig 2B) shows local lipid packing defects near the binding site compared to distant unperturbed regions, indicating membrane remodeling induced by the bound peptide.

Unlike the amphipathic helix (AH) region, the extended residues do not maintain a clear spatial segregation of hydrophobic and hydrophilic side chains. As a result, not all hydrophobic residues on the extended domains can orient toward the membrane lipid tails. These hydrophobic residues that face outward are highlighted by arrows in Fig 2A. This contrasting amphipathic character between the AH and extended domains may contribute to the overall curvature of the bound peptide. The radius of the fitted circle, representing the overall radius of curvature, was found to be approximately 60 Å in all replicas (see Fig A in S1 File). This bending was not observed in our previous simulations of the shorter AH domain comprising only 18 residues [14], which remained largely straight when membrane-bound. The introduction of extended

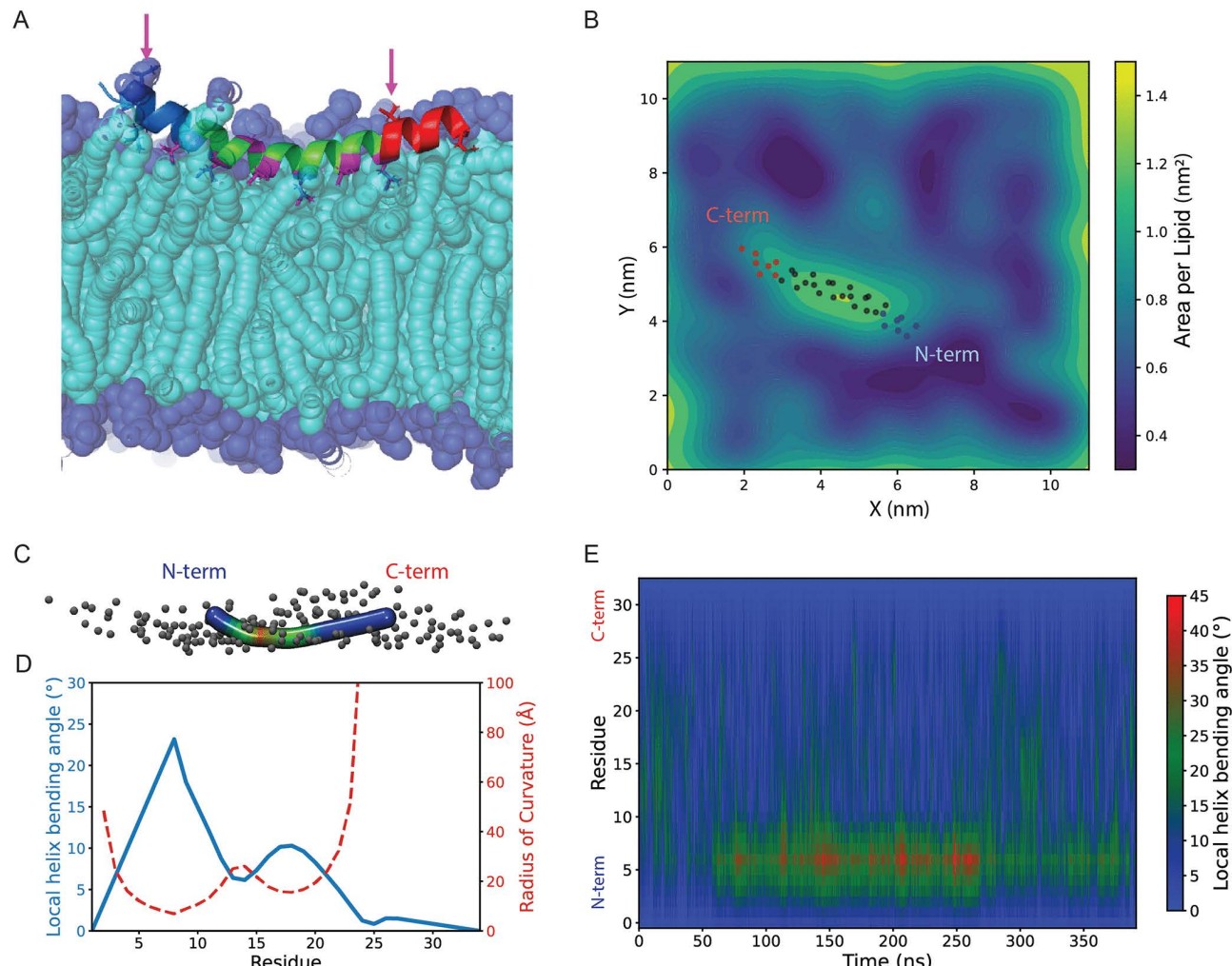

**Fig 2. Bound peptide adopts a stable and smile-like curvature in the membrane. (A)** MD simulation snapshot of a membrane-bound extended AH, with the N-terminus in blue and the C-terminus in red. Arrows indicate hydrophobic residues in the extended regions that face outward toward the solution, unlike the hydrophobic face of the AH region that orients inward. **(B)** Snapshot of the spatial map of area per lipid (APL) in the membrane upper leaflet. The presence of the peptide increases local APL near its binding site. The dots indicate the position of the backbone alpha carbons. **(C)** Peptide conformation color-coded by local bending angle using the Bendix tool. **(D)** Residue-resolved local bending angle (blue, left axis) and corresponding radius of curvature (red dashed, right axis) corresponding to the Bendix snapshot. **(E)** The local helix bending angle map over simulation time shows stability in the local bending of the membrane-bound peptide. N-terminal residues bend more than those on the C-terminal side.

regions, which lack clear amphipathic character, gives rise to structural asymmetry that may contribute to the observed bending within the membrane environment.

Fig 2C shows a representative snapshot of the peptide, color-coded based on the local bending angle in the VMD Bendix tool. The corresponding local bending angle and radius of curvature of the snapshot is shown in Fig 2D. The peptide bending is not evenly distributed: the N-terminal region exhibits greater bending than the C-terminal region, and this asymmetric bending pattern remains stable throughout the simulations. This trend is visualized in the heatmap of the local helix bending angle per residue over time in Fig 2E (also see Fig B for additional replicas in S1 File). Notably, the highest degree of bending in the membrane-bound peptide occurs near residue G7, located at the interface between the N-terminal extension and the start of the AH domain. Since the AH domain spans residues 9–26, this sharp bend at the

boundary likely reflects an interface effect, as further illustrated by the helix bending angle time average profile shown in Fig C in S1 File.

### Inter-peptide interactions modulate the positioning and orientation of the unbound peptide

To test whether a membrane-bound peptide can facilitate the recruitment of additional peptides, we simulated a system containing two extended AH peptides: one initially bound to the membrane and the other initially placed in solution. As a control, we also analyzed a system of a single unbound peptide near the membrane. The Z-component of the center of mass (Z-COM) distribution for the unbound peptide shows a clear peak difference between the positioning of the unbound peptide in the single- (orange) and two-peptide (green) systems in Fig 3A. The gray curve, representing the distribution of phosphate headgroup positions, serves as a membrane reference. In the two-peptide system, the unbound peptide is more likely to reside closer to the membrane, as shown by a higher peak in the Z-COM histogram compared to the single-peptide system. This is further supported by Fig D in S1 File, which shows the AH domain Z-COM position over simulation time for each replica of the single-peptide system ($n = 4$) and the two-peptide system ($n = 8$). In the two-peptide systems, the unbound peptide remains closer to the membrane with reduced vertical fluctuations compared to the single-peptide systems that exhibit more variability in terms of membrane proximity.

To better understand the orientation of the unbound peptide toward the membrane, we examined the Z-COM positions of residues for the unbound peptide in both systems (Fig 3B, C; see also Fig E in S1 File). In the single-peptide system (Fig 3B), the C-terminal region (red) approaches the membrane more closely than the N-terminal region (blue), likely driven by electrostatic attraction between the positively charged C-terminal residues and the negatively charged phosphate headgroups. This preferred approaching direction to the membrane via the C-terminus is also visible in the simulation snapshot (Fig 3A, left inset).

On the other hand, in the two-peptide system (Fig 3C), the N-terminal region—particularly the first four residues—shows closer proximity to the membrane than in the single-peptide case. When there is a bound peptide, both termini of the unbound peptide show proximity to the membrane. The N-terminal shift is likely mediated by direct interaction with the membrane-bound peptide, as shown in the snapshot (Fig 3A, right inset). Notably, three of the first four N-terminal residues are charged (arginine and glutamic acid), suggesting a possible role for electrostatic interactions in mediating inter-peptide contacts and reorienting the unbound peptide.

To further investigate the floating peptide-lipid interactions, we performed a residue-resolved contact analysis for PLPI and DOPC headgroups (Fig F in S1 File). Both lipids display elevated contact levels near the peptide termini. However, PLPI exhibits additional mid-sequence interactions that are more pronounced than those observed with DOPC. Contacts are strongest at cationic residues, with early Arg positions and the C-terminal Lys-rich segment displaying the highest interaction levels. Charged contacts are also observed for DOPC due to its zwitterionic headgroup. We note that DOPC is approximately threefold more abundant than PLPI in our membrane composition.

### Charged residues form inter-peptide contacts in a preferred anti-parallel orientation

In the two-peptide simulations, although both termini of the unbound peptide occasionally come into proximity of the membrane surface, the N-terminus tends to remain closer to the bound peptide, whereas the C-terminus more often faces the membrane, engaging with phosphate headgroups. These observations raise the question of whether stable inter-peptide contacts form and whether specific conserved residues (see Fig 6B)—particularly charged ones—play a role in mediating these interactions.

To examine inter-peptide interactions, we computed contact maps based on distances between residues of the two peptides (Fig 4A). We applied a 7.5 Å cutoff to the Cα distances to accommodate interactions mediated by long or flexible side chains. A zoomed-in view highlights a frequent contact between R4 of the floating peptide and E21 of the bound

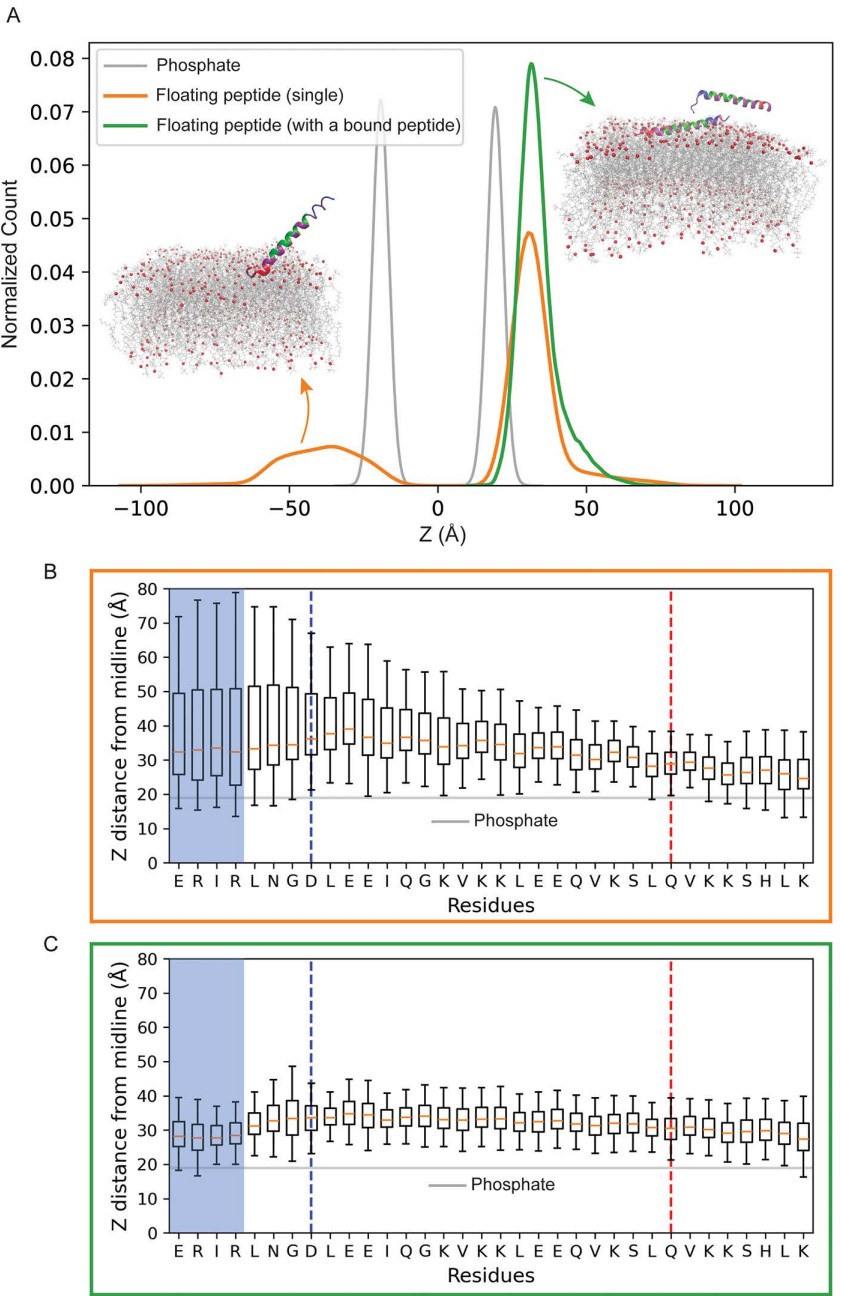

**Fig 3. The presence of a membrane-bound peptide modulates the positioning and orientation of the unbound peptide toward the membrane.**
**(A)** Distribution of the Z-component of the center of mass (COM) for the unbound peptide in systems with single (orange, n = 4) and two peptides (green, n = 4). The gray curve shows the distribution of Z-positions of lipid phosphate headgroups. Left inset: MD simulation snapshot showing the single unbound peptide approaches the membrane via the C-terminus (orange). Right inset: MD simulation snapshot showing an unbound and a bound peptide. The N-terminal region of the unbound peptide can interact with the bound peptide (green). **(B)** Box plot of the Z-COM of residues in the system with a single unbound peptide (n = 4). The blue and red dashed lines separate the N- and C-terminal regions, respectively. **(C)** Box plot of the Z-COM of residues of the unbound peptide in the multiple-peptide system. The first four N-terminal residues (blue box) show a closer approach to the membrane, consistent with interaction with the membrane-bound peptide.

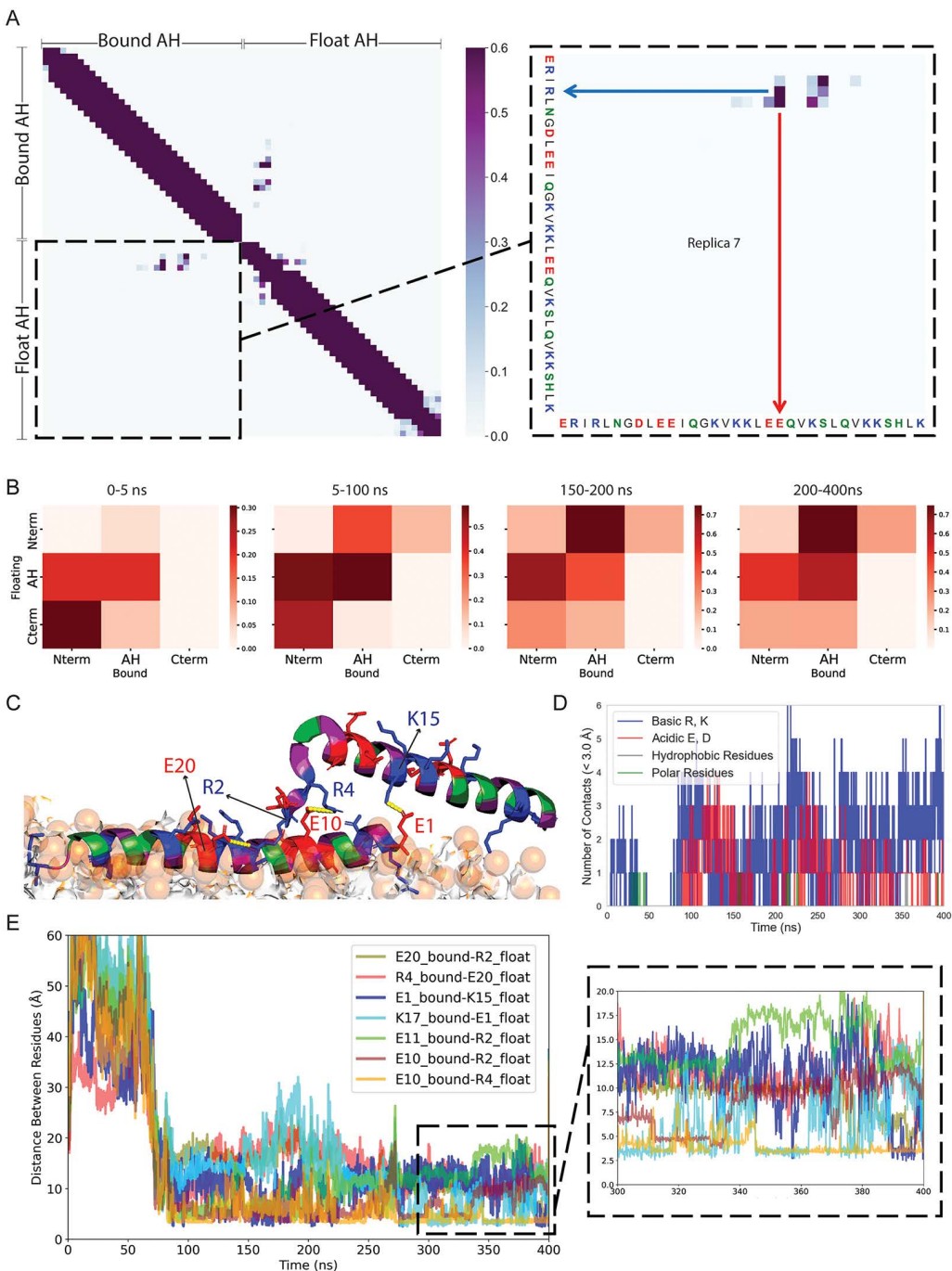

**Fig 4. Salt bridge formation between charged amino acids promotes peptide-peptide interactions. (A)** Contact map showing intra- and inter-peptide contacts. The zoomed-in inter-peptide region highlights a salt bridge between charged residues R4 and E21. The N-terminal of the floating peptide preferentially interacts with the ending region of the AH domain of the bound peptide, indicating an antiparallel peptide orientation. **(B)** The domain-based coarse-grained contact maps averaged over all replicas in different time windows show two peptides interacting in an anti-parallel configuration that reaches a steady state. **(C)** Simulation snapshot showing multiple salt bridges formed between charged residues of the floating and bound peptide. Basic residues are shown in blue, acidic in red, polar in green, and hydrophobic in magenta. (D) number of inter-peptide contacts between residue pairs of the floating and bound peptides. The majority of contacts involve charged residues. **(E)** Time evolution of residue pair distances for the top-ranked salt bridge contacts, revealing how the two peptides interact (replica 2). R and E residues form a stable salt bridge throughout the simulation.

peptide. Fig G in S1 File shows the contact map for other interacting replicas (replica 2, 4, and 5) demonstrating salt-bridge formation between charged arginine residues in the N-terminal region and acidic residues on the other peptide, consistent with those observed in Replica 7 (Fig 4A). Fig 4B presents a domain-level contact map, coarse-grained across defined peptide regions and segmented into time windows for all replicas. The contact maps reveal that the N-terminal region of the floating peptide consistently tends to interact with the AH domain of the membrane-bound peptide. A closer look at the early time windows reveals that the initial dominant interaction occurs between the N-terminus of the bound peptide and the C-terminus of the unbound peptide, forming a preferred anti-parallel orientation. As the simulation progresses, the contact pattern stabilizes, with the most frequent interaction shifting to one between the N-terminus of the unbound peptide and the AH domain of the bound peptide. These interactions often involve charged residues in the N-terminus and within the AH domain, indicating a likely contribution of electrostatic interactions and potential salt bridge formation in stabilizing the anti-parallel alignment of the two peptides.

To assess whether the relative arrangement of the bound and floating AH is robust across initial conditions, we quantified inter-peptide contacts and orientation over eight independent replicas (Fig H in S1 File). We operationally define an "antiparallel-like" interaction as N-terminal-mediated association, where the N-terminal patch (residues 1–8) of either peptide forms heavy-atom contacts (<4.5 Å) with the second half of the other peptide (residues 17–34), and a "parallel-like" interaction as the analogous C-terminal-mediated association, where the C-terminal patch (residues 27–34) contacts the second half of the other peptide. Across replicas, N-terminal-mediated contacts are frequent and long-lived, whereas C-terminal-mediated contacts are rare and typically transient (Fig H in S1 File panels A–B). This N-terminal preference is consistent with the structural trends observed in Fig 4A and Fig G in S1 File, which also show that association events are predominantly initiated by approach and engagement of an AH N-terminus rather than its C-terminus. To quantify peptide–peptide relative orientation, we defined a peptide axis vector v for each peptide as the center-of-mass (COM) vector from the N-terminal patch (residues 1–8) to the C-terminal patch (residues 27–34), using heavy atoms only. For every trajectory frame, we computed $\cos(\theta)$, where $\theta$ is the angle between the two N to C axes thus $\cos(\theta) \approx -1$ corresponds to antiparallel alignment and $\cos(\theta) \approx +1$ to parallel alignment. We further classified frames as "interacting" if the inter-peptide contact count exceeded a minimal threshold ($T > 5$) over 8 replicas, where A counts contacts between the N-terminal patch of one peptide and the second half of the other peptide (residues 17–34) and P counts contacts between the C-terminal patch and the second half, using a 4.5 Å heavy-atom cutoff. We then compared the orientation distributions $\cos(\theta)$ for interacting versus non-interacting frames pooled across replicas. The resulting distributions show that peptide orientation is strongly state-dependent: interacting frames are enriched at $\cos(\theta) \approx -1$, indicating that when the peptides form direct contacts, they preferentially adopt antiparallel-like alignment. In contrast, non-interacting frames are broader, consistent with peptides sampling a wide range of relative orientations when separated. Consistent with this contact-based definition, frames classified as interacting are enriched in antiparallel-like orientations (negative $\cos \theta$), while non-interacting frames sample a broader range of relative orientations (Fig H in S1 File panel C). Together, these results indicate that the dominant association mode is reproducible across replicas and is biased toward N-terminal engagement of the partner's C-terminal half rather than a parallel, C-terminal-mediated arrangement.

Salt bridges—electrostatic interactions between oppositely charged residues—are known to play a key role in stabilizing protein-protein interactions and overall structural integrity [24]. As illustrated in the simulation snapshot (Fig 4C), multiple inter-peptide salt bridges form, primarily involving the N-terminal region of one peptide contacting the AH domain of the other.

We quantified inter-peptide contacts by residue class (acidic, basic, polar, and hydrophobic) using heavy-atom distances with a 3.5 Å cutoff evaluated at each frame. As shown in Fig 4D for replica 2 (see Fig I in S1 File for all replicas), contacts involving charged residues dominate over polar or hydrophobic residues in forming close inter-peptide contacts.

To further characterize these charged interactions, we tracked the top-ranked residue pairs using time-evolution traces from the VMD salt-bridge analysis (Fig 4E for replica 2 and Fig J in S1 File for other replicas). These diagrams indicate that interactions between arginine (R) and glutamic acid (E) are recurrent throughout the simulation trajectories. The occupancy

analysis (Fig J in S1 File panel B and Table A in S1 File) shows that salt bridges are intermittent and differ across replicas, as reflected by the standard deviations. Nonetheless, the overall salt-bridge signal is dominated by a small subset of residue pairs. This enrichment is particularly clear for arginine: although the peptide contains only two Arg residues (R2 and R4), several of the most populated contacts involve Arg (R4–E10, R4–E21, R4–E20, R2–E10), indicating a disproportionate contribution of Arg-mediated interactions during AH–AH association. The lifetime analysis (Table B in S1 File) further shows that these interactions recur as multi-frame events rather than isolated single-frame contacts. K18–E10 and R4–E10 are the most recurrent pairs, with 40–50 events and the largest total on-times (321 ns and 280 ns). In contrast, R4–E20 occurs less frequently but can form longer-lived episodes (median 18 ns; 95th percentile 73.8 ns). Together, these occupancy and lifetime statistics provide a quantitative foundation for the charged interactions highlighted in the manuscript.

We also investigated the structural behavior of tandem AH peptides in which two AH domains were connected by a 10-residue linker introduced in an experimental study [15]. In this experimental construct, the C-terminus of the first AH domain is linked to the N-terminus of the second AH domain, forming a structure we refer to as a tandem NC-NC or anti-parallel AH domains. We performed all-atom simulations of this 46-residue tandem construct initially unbound. Although the simulation started with a straight conformation, the peptide gradually bent at the linker region, resulting in an anti-parallel arrangement of the two AH domains (Fig K in S1 File panel A). A positively charged arginine in the linker interacted with the membrane surface, anchoring the peptide near the bilayer without full insertion. In this anti-parallel conformation, the two AH domains remained folded, with their hydrophobic faces aligned and helical content preserved throughout the simulation (Fig K in S1 File panel D). In contrast, when the first AH domain is flipped to form a CN–NC tandem configuration in which the peptides adopt a parallel arrangement, the structure is less stable: the helices unfold, and therefore, the hydrophobic and hydrophilic faces are no longer clearly maintained (Fig K in S1 File panel B). The contact map and helical content diagram (Fig K in S1 File panel E) show lower helicity and random contacts in the parallel CN–NC case.

To test whether structural stability could be restored by mimicking more native septin features, we extended both AH domains with eight additional C-terminal residues from Cdc12, creating an "extended parallel" tandem peptide. In this system, both C-termini contributed to membrane interactions. Unlike the shorter parallel tandem, this construct retained helical and amphipathic structures throughout the simulation (Fig K in S1 File panels C, F), likely due to the stabilizing influence of the charged C-terminal extensions. These results support our previous findings [14] that the addition of flanking regions enhanced the helicity and stability of the structure. Tandem simulation results suggest that both anti-parallel geometry and the presence of extended, charged termini contribute to the structural integrity of tandem AH domains. Notably, the anti-parallel configuration observed in these tandem simulations reinforces the stable orientation of the anti-parallel alignment observed in our two-peptide simulations. Fig L in S1 File shows the last frame replicas for all tandem simulations to visually show the stability of the helices.

## The positioning of bound peptides in the membrane and the organization of the lipids depend on the number of bound peptides

We further explored peptide–peptide interactions by analyzing systems containing two peptides initially bound and shallowly inserted into the lipid bilayer in a parallel conformation that the previous section showed is less favorable. We intentionally initialized the two peptides in a parallel arrangement, despite its lower stability relative to the anti-parallel configuration, to provide a conservative test of cooperative binding. The persistence of peptide–peptide association even in this unfavorable geometry underscores the strength of AH–AH interactions and supports their potential role in septin assembly. Notably, both our previous atomistic simulations and complementary coarse-grained simulations favor anti-parallel arrangements, suggesting that the interactions observed here likely represent a lower bound on cooperative stability.

Across eight simulation replicas, we observed that in 75% of the cases, both peptides remained bound to the membrane while maintaining contact with each other (Fig 5A). In the remaining 25%, one peptide partially dissociated from

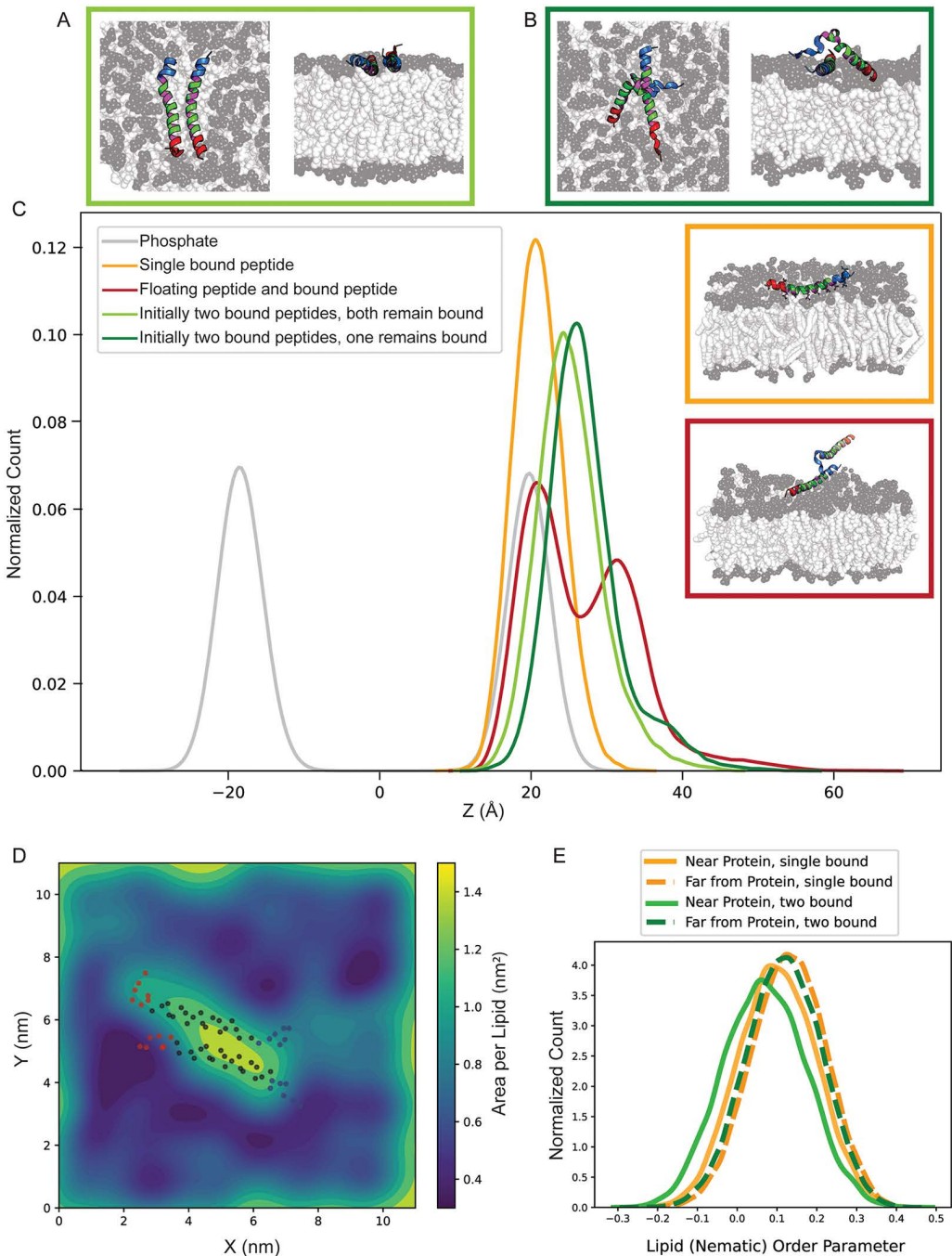

**Fig 5. The position of bound peptides and the lipids arrangement change in single- and multiple-peptide systems. (A and B)** Top and side view simulation snapshots of systems with the same initial structure of two parallel bound peptides show different behavior: (A) Both peptides remain bound in 400 ns simulation time (light green, 75% of replicas) **(B)** One of the peptides dissociates from the bilayer and interacts with the other peptide. (dark green, 25% of replicas). **(C)** Z-component of the center of mass distribution of the bound peptide in different systems. Inset: Simulation of snapshots of a single membrane-bound peptide (orange) and a system of bound and unbound peptides with lipid bilayer (red). **(D)** Spatial map of area per lipid (APL) in the membrane upper leaflet for two bound peptides initially placed in parallel (the non-favorable) orientation. The two peptides increase the local APL near the binding site and compact the distant lipids. The dots indicate the position of the backbone alpha carbons with N-terminal residues in blue and C-terminal residues in red. **(E)** Histogram of the lipids order parameter for systems with a single (orange) and two (green) bound peptides in the vicinity of (solid line) and far from (dashed line) the binding site.

the bilayer and shifted toward the other peptide, forming interactions primarily through its N-terminal region (Fig 5B). The snapshot's side and top views demonstrate how one peptide reorients away from the membrane to associate with another peptide.

In our single-peptide simulations, the bound state remained stable over the sampled timescales, with no evidence of spontaneous partial detachment. In contrast, in 25% of two-peptide systems, the presence of a second peptide— even when initialized in the less favorable parallel orientation— was accompanied by partial detachment of one peptide from the membrane. This observation suggests that inter-peptide interactions can, in some cases, compete with peptide–membrane associations. These outcomes imply that peptide–peptide contacts may provide a stabilizing influence that reshapes membrane binding dynamics in multi-peptide systems. Because these trajectories are 400 ns long, we interpret this as a kinetic outcome observed within the sampled window rather than a converged dissociation probability.

To provide an overview of peptide positioning relative to the membrane, we compared the Z-component of the center of mass distributions across different systems (Fig 5C). In systems with two bound peptides (green), the Z-COM peak shifts to the right relative to that of the single-peptide system (orange), indicating that two bound peptides tend to reside slightly upward in the membrane. This upward shift becomes more pronounced over time, as shown in the time-resolved Z-COM distributions in Fig M in S1 File, suggesting the gradual displacement of the peptides during the simulation.

One possible explanation for this shift is lipid remodeling induced by increased peptide occupancy. The area per lipid (APL) map (Fig 5D) reveals local lipid packing defects: APL values are higher near the peptides, while more distant regions become increasingly compacted. Compared to the more uniform lipid distribution in the single-peptide system (Fig 2C), the two-peptide system displays more pronounced spatial heterogeneity, with more dark blue regions reflecting increased lipid density far from the binding site. This localized crowding, combined with inter-peptide interactions, may generate lateral pressure that slightly displaces the peptides from their original membrane-embedded positions. Consistent with this, the lipid order parameter distribution (Fig 5E) shows that lipid tails become increasingly disordered as the number of bound peptides increases, particularly near the binding sites. The reduction in the order parameter likely reflects local perturbations of membrane structure caused by multiple interacting peptides. Together, these results suggest that increasing peptide occupancy alters both membrane organization and peptide positioning. In this context, a more curved membrane—by increasing available lipid area and reducing packing constraints—may better accommodate multiple bound peptides.

To further assess whether the antiparallel, electrostatically stabilized AH–AH interaction motif identified in two-peptide systems persist beyond a peptide pair, we performed additional simulations containing three AH peptides. These simulations were initialized by introducing a third peptide into systems in which a bound peptide and a second peptide had already formed a stable antiparallel interaction at the membrane interface. Across multiple independent replicas, the third peptide engaged the existing assembly through similar antiparallel orientations and stable interactions, indicating that the interaction motif can extend beyond a single peptide pair.

Notably, in one replica, the third peptide exhibited enhanced membrane engagement when it is distant from the pre-assembled pair, including insertion of its C-terminal region into the lipid headgroup region (Fig N in S1 File panel B). This C-terminal insertion was not observed in single-peptide simulations. This observation suggests that the presence of multiple membrane-associated AHs can cooperatively modify the local membrane environment and lower the energetic barrier for peptide–membrane association. Representative snapshots illustrating the temporal evolution of these three-peptide systems are shown in Fig N in S1 File. While these simulations do not yet constitute filament-scale assemblies, they provide direct evidence that the proposed antiparallel AH interaction motif can be propagated in small multivalent systems and support the idea that cooperative AH–AH and AH–lipid interactions may underlie higher-order septin assembly. A more systematic exploration of higher-order assembly will be an important direction for future work.

## Discussions and conclusion

Our simulations suggest a mechanistic model for how septin AH domains interact to form assemblies on curved membranes. As illustrated in Fig 6A, a membrane-bound septin may facilitate the recruitment and alignment of neighboring septin complexes through salt bridge interactions between oppositely charged residues on their amphipathic helices. These interactions stabilize a preferred anti-parallel arrangement between AH domains, allowing for cooperative membrane association while preserving peptide helicity and amphipathic character. This configuration may represent an early step in septin filament formation, where lateral peptide–peptide contacts, stabilized by electrostatics, promote filament growth along the membrane surface. While our simulations were performed on planar bilayers, the interaction motifs identified here are consistent with models in which membrane curvature—by increasing lipid spacing and reducing local packing density—could further favor cooperative AH engagement.

Antiparallel packing of amphipathic α-helices stabilized by electrostatic complementarity has precedents in other lipid-associated assemblies. A well-characterized example is provided by exchangeable apolipoproteins, where amphipathic helices adopt "belt"-like organizations around discoidal lipoprotein particles; in particular, detailed modeling of apoA-I supports an antiparallel helix–helix interface and explicitly implicates networks of intermolecular salt bridges in selecting and stabilizing preferred registries of the antiparallel dimer [25,26]. Recent structural work on lipidated ApoE likewise supports an antiparallel dimer organization on nascent discoidal HDL-like particles [27]. Related antiparallel helical assemblies have also been proposed in membrane-active amphipathic peptides, where electrostatic interactions can contribute

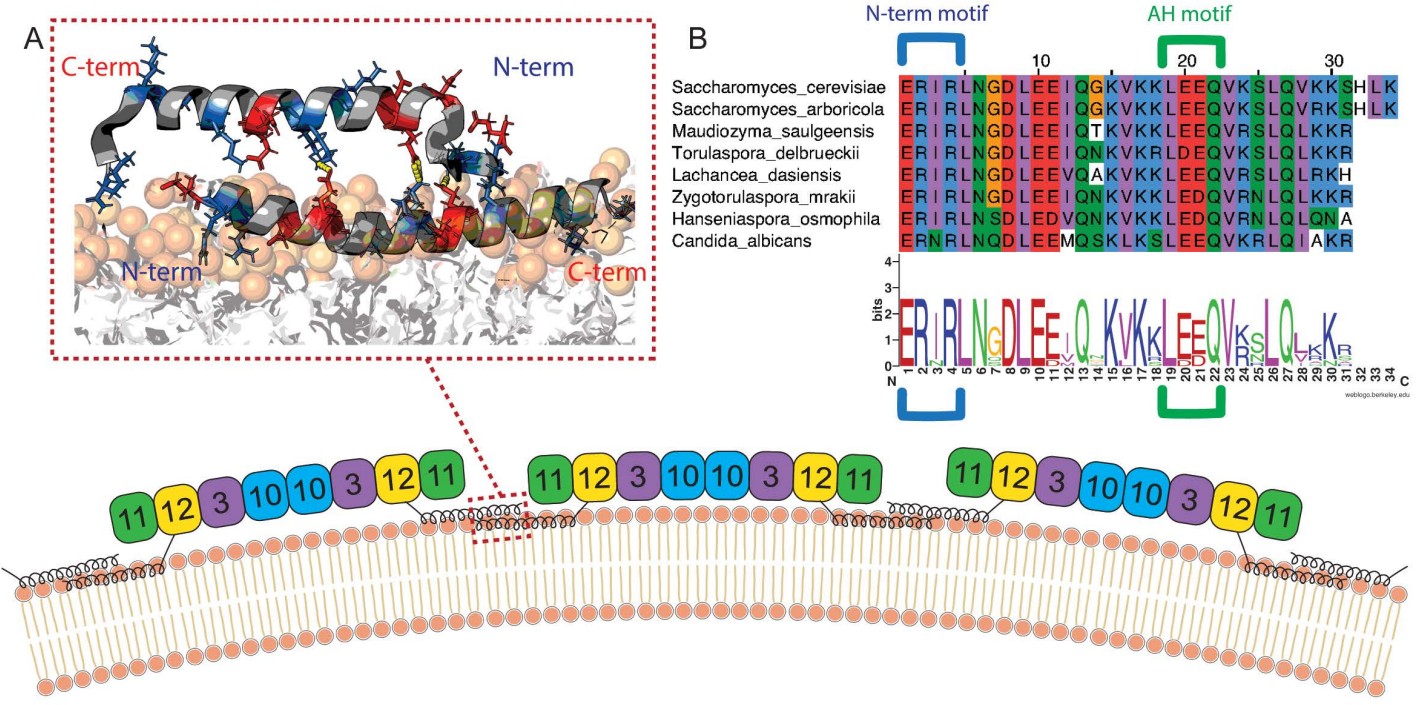

**Fig 6. Proposed mechanism of septin polymerization on the curved lipid bilayer. (A)** The bound septin facilitates the binding of neighboring septins through salt bridge interactions between their AH domains in anti-parallel, forming septin assemblies. **(B)** Multiple sequence alignment and sequence logo of the Cdc12 amphipathic helix domain across fungal species. Conserved charged residues implicated in inter-peptide salt bridges are highlighted, including arginine residues in the N-terminal **ERIR** motif and glutamic acid residues in the **LEE** motif. These charged residues are instrumental to salt bridge interactions and peptide-peptide contacts.

to complex formation alongside other determinants [28]. Together, these examples suggest that antiparallel AH–AH association can be a recurring solution for building electrostatically complementary interfaces at lipid surfaces, but it is not necessarily a universal motif. The emergence and stability of antiparallel packing are expected to depend on charge patterning, helix length/helicity, and the broader protein architecture that constrains AH orientation. In the case of septins, the conserved distribution of charged residues within the AH and flanking regions appears well suited to support electrostatically stabilized antiparallel interactions in a membrane-associated context, thereby promoting cooperative engagement.

Importantly, the charged residues involved in these stabilizing interactions are evolutionarily conserved across septin homologs. Fig 6B presents a sequence alignment across species, emphasizing the conservation of key charged residues. Glutamic acid residues involved in salt bridge formation is located within the conserved LEEQV motif of the Cdc12 AH domain, while the interacting arginine in the N-terminal extension ERIR motif is also conserved across species [15,29,30]. These conserved features may point to a universal strategy used by septins to stabilize their membrane-bound assemblies. Experimental validation through targeted mutagenesis of these residues would offer a valuable test of their role in inter-peptide interaction and filament formation.

Simulations of a single membrane-bound extended AH peptide revealed a stable smile-like curved conformation. This bending behavior is not observed in shorter constructs [14]. This curvature is driven by the inclusion of non-amphipathic flanking regions that introduce asymmetry and enhance bending flexibility. Local curvature analysis revealed that the bending is asymmetric and most pronounced near the N-terminal interface. When a second peptide was added to the solution, its N-terminal region consistently approached the membrane-bound partner and formed inter-peptide interactions. In particular, salt bridges are frequently formed between arginines in the N-terminal extension of the floating peptide and glutamic acids within the AH domain of the membrane-bound peptide. These interactions potentially stabilize an anti-parallel configuration between the two peptides and suggest a mechanism of cooperative membrane association, where one bound peptide facilitates the recruitment and stabilization of another, reorienting and stabilizing it at the membrane interface. Because the AH domains are modeled in isolation, the N-terminal region of the floating peptide displays partial unfolding; in full-length Cdc12, this region would be structurally stabilized, which may modulate the peptide–peptide interactions observed here. We note this as a limitation of the isolated-peptide model. Also, it is important to note that, while changes in lipid composition or ionic strength would be expected to modulate the balance between electrostatic and hydrophobic contributions to AH binding, the cooperative interaction motifs identified here arise from local peptide–peptide and peptide–lipid interactions and are therefore expected to persist qualitatively across a range of physiologically relevant conditions. We also note that both termini of the extended AH constructs were neutrally capped in the simulations. In full-length Cdc12, the C-terminus of this region would carry a negative charge. Capping would alter the net charge of the C-terminal segment by one unit. While this difference may modestly alter the strength of electrostatic interactions, it is unlikely to qualitatively affect the interaction motifs identified here, which are driven by distributed charge patterns and conserved residue-specific salt bridges rather than by the terminal charge alone.

Simulations of tandem AH constructs provided additional insights: peptides with NC–NC linkage adopted a folded, anti-parallel configuration stabilized by inter-domain contacts, while CN–NC tandem constructs showed partial unfolding and reduced stability. Interestingly, extending AH domains with C-terminal residues derived from Cdc12 restored stability even in parallel arrangements. This result highlights the role of terminal extensions in preserving helicity and amphipathic structure.

We also observed that increasing the number of bound peptides leads to structural changes in both peptide positioning and membrane organization. Two bound peptides induced local packing defects and lipid tail disorder, creating spatial heterogeneity in the membrane that differs from the single-peptide case. The area per lipid maps and lipid order parameters revealed lipid compaction at distal regions and increased disorder near the binding sites, which may generate lateral pressure that subtly displaces peptides from the membrane interior. This suggests that cooperative peptide binding may not only rely on inter-peptide contacts but also emerge from lipid-mediated effects that reorganize the membrane to accommodate multiple peptides. Such lipid-mediated accommodation is expected to be further enhanced in curved membranes, where packing defects are intrinsically more abundant, although this remains to be tested explicitly.

 

While our results highlight a correlation between increased peptide occupancy, lipid tail disorder, and subtle peptide displacement, we categorize these functional interpretations—such as the role of lateral pressure or specific defect compatibility as preliminary hypotheses. Future studies should directly probe the effects of membrane curvature using explicitly curved geometries to validate these observations. Furthermore, to move beyond qualitative correlations, additional rigorous analyses, including the calculation of lateral pressure profiles and free-energy landscapes will be essential to provide a thermodynamic basis for the displacement and recruitment behaviors reported here. Such efforts, combined with enhanced sampling techniques to observe spontaneous binding and multivalent assemblies on longer timescales, will refine our understanding of how septins sense and stabilize cellular shape. Such studies may also inform the rational design of synthetic peptides and materials capable of mimicking biological curvature sensing and cooperative assembly mechanisms.

In summary, our findings suggest that the presence of a membrane-bound peptide enhances the membrane proximity of the unbound peptide, possibly by creating a local interaction site or perturbing the lipid environment to favor association. These findings highlight the importance of both sequence features and membrane context in shaping AH domain interactions and suggest a plausible molecular mechanism by which septins assemble into filaments on the lipid bilayer.

## Materials and methods

This computational study of a system with peptides and lipid bilayer is performed using GROMACS 2022.3 [31]. The amino-acid sequences were derived from the septin protein in budding yeast S. cerevisiae Cdc12, which contains the amphipathic helix (AH) domain [15]. We prepared the initial structure of the peptides using Pymol. Then the complete system, including the peptide, lipid bilayer, and the solution of water and ions, was assembled using the CHARMM-GUI web server [32] utilizing the CHARMM36m force field [33,34]. The salt concentration in the solution is 50 mM KCl, and the temperature is 300 K. We set a timestep of 2 fs, with Van der Waals (VDW) interactions switched between 1.0 and 1.2 nm, and particle mesh Ewald was used at long distances to compute the electrostatic interactions [35]. We used the LINCS algorithm to constrain hydrogen bond lengths [36]. Evolution of the system was performed by the Nose-Hoover thermostat with a coupling constant ($\tau t$) of 1 ps and Parrinello-Rahman barostat at a reference pressure of 1.0 bar and compressibility of $4.5 \times 10 - 5$ with a coupling constant of ($\tau p$) of 5 ps. The pressure was coupled semi-isotropically in XY for membrane simulations and isotropically for peptide-only simulations.

All simulations were performed on membrane patches of approximately 11 × 11 nm². This system size was chosen to balance computational feasibility with sufficient lateral separation between peptides and their periodic images. The extended AH peptides span approximately 4–5 nm, leaving several nanometers of intervening lipid between peptides and box boundaries. Previous computational studies have shown that lipid packing defects and acyl-chain perturbations induced by amphipathic helices are highly localized around the insertion site [37–39]. Accordingly, the present system size is sufficient to capture local peptide-induced membrane reorganization while minimizing direct interactions with periodic images.

### Single extended AH domain

The extended AHs, including the AH domain in the middle (18 residues) and extended residues on both ends (8 residues on each side) create a 34-residue peptide we call extended AH (*ERIRLNGDLEEIQGKVKKLEEQVKSLQVKKSHLK*). This sequence is located on the C-terminal of S. cerevisiae Cdc12 (Fig 1). The net charge of the middle AH domain and eight upstream amino acids on the N-terminal are neutral, but the eight downstream amino acid sequence on the C-terminal has a + 3 net charge under our experimental conditions (pH = 7.4). First, we build an idealized $\alpha$ helix of these 34 amino acids in PyMOL [40] and utilize CHARMM GUI to add neutral cappings for both N-terminal (NNUE) and C-terminal (CT2). The use of the CT2 neutral cap removes the native −1 charge at the C-terminus, making the AH construct one unit more positively charged than the physiological uncapped state. We note this as a minor limitation of the model, although this

small difference in net charge is not expected to affect the qualitative behavior observed in our simulations. Previous work has shown that terminal capping can also alter local electrostatics and helicity in ways that influence membrane-binding energetics [14].

### Lipid bilayer

To build the lipid bilayer, we utilized the CHARMM-GUI membrane builder [41,42] (Jo et al., 2009; Wu et al., 2014). All simulations employed a DOPC:PLPI 75:25 bilayer composition. This lipid ratio was chosen to ensure consistency with our previous computational study [14], facilitating direct comparison between single- and multi-peptide behaviors. The same lipid mixture has also been used experimentally in septin–membrane binding assays [15], providing additional support for its relevance as a model membrane environment. The XY size of the simulation box is set to be roughly $11 \times 11 \, nm^2$ and solvated with at least 5 nm of water (TIP3P forcefield) above and below the bilayer. The extended AHs are aligned with the bilayer in CHARMM-GUI so that the ILE12 and LEU19 residues are oriented to face downward to the interior, the hydrophobic portion of the bilayer (Fig 1B and 1C). For the bound initial structure, we create a system of the lipid bilayer and place the extended AH parallel and at a z = 15 Å distance to the membrane midline. For the unbound initial structure, the distance of the extended AH center of mass from the midline is z = 30 Å. Systems were relaxed and equilibrated using the CHARMM-GUI steps [42].

### Two peptides

**An unbound and a bound extended AH on the lipid bilayer.** We create a system of the lipid bilayer with a bound extended AH and an extended AH unbound in the solution above the bilayer: to build this, we took the last frame of single-bound peptide simulations and added another extended AH to the solution. The unbound peptide is located parallel at a 30 Å distance from the bound peptide. Then, we deleted overlapping water and ion atoms within a distance of 7 Å of the unbound peptide atoms. In this method, the bound peptide is already in the bending shape with some hydrophobic residues on the extended parts facing outward of the bilayer.

**Two bound extended AHs on the bilayer.** The initial structure includes two bound extended AHs in the parallel conformation (C-terminal of both in one direction) with a 15 Å distance relative to each other. We placed these two parallel peptides in a box of lipid bilayer, water, and ions using CHARMM GUI. Again, we align ILE12 and LEU19 orientation so that the hydrophobic residues of the AH domain face the hydrophobic lipid tails and hydrophilic residues face the polar lipid phosphate headgroups—both peptides at the height of z = 15 Å from the bilayer midline.

### Tandem AH domains

We connected two AH domains with the 10 amino acid sequence described in the experimental study [15]. We generated three types of tandem structures:

1) **Tandem NC-NC AH domains:** linking the C-terminus of the first AH domain to the N-terminus of the second one. In this way, we get a 46-residue sequence called tandem AH NC-NC: *LEEIQGKVKKLEEQVKSLGSGSRSGSGSLEEIQGKVKKLEEQVKSL.*

2) **Tandem CN-NC AH domains:** linking the N-terminal of the first peptide to the N-terminal of the second AH domain. In other words, we flipped one of the AH domains to form a peptide with 46 amino acids, that only the C-termini of the AH domains are on the ends. We call this structure tandem AH CN-NC, or a symmetric tandem that is formed by a palindromic arrangement of AH domains: *LSKVQEELKKVKGQIEELGSGSRSGSGSLEEIQGKVKKLEEQVKSL.*

3) **Tandem extended CN-NC AHs:** The palindromic arrangement of AHs plus the extended charged C-termini on both ends. It forms the sequence of 62 amino acids:

*KLHSKKVQLSKVQEELKKVKGQIEELGSGSRSGSGSLEEIQGKVKKLEEQVKSLQVKKSHLK*. This design can be considered as a molecule that can mimic the septin protein structure with two AH domains and eight extended C-terminal residues on the two Cdc12s, regardless of the middle residue sequences and their length.

### Three peptides

To construct three-peptide systems, we extended the two-peptide setup by introducing a third extended AH peptide into the simulation box. Specifically, we selected the final frame of two-peptide simulations in which a bound and an unbound peptide had formed a stable antiparallel interaction at the membrane interface. A third extended AH peptide was then placed in the solution above the bilayer, initially positioned parallel to the membrane surface and separated from the existing peptides.

**Post-processing and analysis of the simulation outputs** A summary of molecular dynamics simulations can be found in Table 1. We analyzed the simulation trajectories using Python scripting and packages of MDAnalysis [43], MDtraj [44], and membrane-curvature [45]. Visualizations were done by PyMOL and VMD [46].

Salt bridge interactions between peptide residues were analyzed using both VMD and PyMOL tools. VMD's salt bridge plugin was used for trajectory-based analysis of inter-residue electrostatic contacts, while PyMOL's salt bridge analysis was applied to representative structural snapshots for visualization and verification.

To assess conservation of charged residues involved in inter-peptide interactions, a multiple sequence alignment of the Cdc12 amphipathic helix domain was constructed using Cdc12 orthologs from representative fungal species. Amino acid sequences of the extended amphipathic helix (AH) domains from selected species were retrieved from the UniProt database [30]. Multiple sequence alignment was performed using Jalview [47], and the aligned sequences were visualized with WebLogo [48] to highlight conserved positions. Charged residues were specifically examined to assess conservation patterns relevant to inter-peptide interactions and salt bridge formation.

To analyze the local bending of the peptide backbone, we used the Bendix plugin within the VMD package [49] to compute and visualize local bending angles, quantitative measures of how sharply the peptide backbone deviates from linearity at specific points along its contour, which are closely related to local curvature. In this analysis, we set the side parameter in Bendix to 7.2 Å, reflecting the chord length measured between pairs of Cα atoms along the helical backbone to define each segment for curvature calculation. This corresponds approximately to the straight-line distance between Cα atoms separated by ~3–4 residues in an α-helix, effectively covering a full helical turn. This setting balances resolution and smoothness in the curvature profile: smaller values increase local sensitivity but may amplify thermal noise, whereas larger values average out finer structural features. Because our analysis focuses on qualitative features of the bending profile, such as asymmetric curvature and the location of maximal bending, moderate variations in this parameter are not expected to qualitatively alter the observed trends. To quantify the overall curvature, we fitted a circle to the backbone

**Table 1. Summary of molecular dynamics simulations.**

| Simulation setup | Replicas, duration (ns) | Peptide(s), cappings |
| --- | --- | --- |
| Single unbound | 4 × 400 | Extended AH, NNUE, CT2 |
| Single bound | 4 × 400 | Extended AH, NNUE, CT2 |
| Unbound and bound 30 Å distance | 8 × 400 | Extended AH, NNUE, CT2 |
| Both bound, 15 Å distance | 8 × 400 | Extended AH, NNUE, CT2 |
| Tandem NC-NC, unbound | 4 × 400 | Linked AH domains, NNUE, CT2 |
| Tandem CN-NC, unbound | 4 × 400 | Linked AH domains, CT2, CT2 |
| Tandem extended CN-NC, unbound | 4 × 400 | Linked Extended AH, CT2, CT2 |
| Two unbound, one bound | 3 × 1000 | Extended AH, NNUE, CT2 |

alpha carbon positions of the peptide projected onto its best-fit plane. This plane was determined using principal component analysis (PCA).

For lipid organization analysis, we used the FatSlim package [50] to compute two key structural properties of the membrane: the area per lipid (APL), and the nematic order parameter of lipid tails. We generated maps of APL across the membrane plane in the form of two-dimensional heatmaps highlighting local variations in lipid packing density. These maps help visualize regions of tight or loose lipid organization that may be associated with protein insertion effects.

To assess the alignment of the lipid tails, we calculated the nematic order parameter, which quantifies the degree of orientational ordering of hydrocarbon tails relative to the membrane normal. For each lipid, the nematic order is defined as the $S = < \frac{3cos\theta^2 - 1}{2} >$, where $\theta$ is the angle between a carbon-carbon (C–C) bond vector along the lipid tail and the membrane normal vector. The angle brackets denote averaging over all bond vectors within a single lipid. This approach enables us to resolve local variations in membrane ordering by capturing differences in lipid tail orientation across the bilayer plane.

## Supporting information

**S1 File. Supporting material.**
(PDF)

## Acknowledgments

We would like to thank Pilar Cossio, Brandy N. Curtis, and Ellysa J. D. Vogt for their helpful discussions. The Flatiron Institute is a division of the Simons Foundation.

## Author contributions

**Conceptualization:** Christopher J. Edelmaier, Ehssan Nazockdast, Sonya M. Hanson.

**Data curation:** S. Mahsa Mofidi, Abhilash Sahoo.

**Formal analysis:** S. Mahsa Mofidi, Abhilash Sahoo.

**Funding acquisition:** Ronit Freeman, Sonya M. Hanson.

**Methodology:** S. Mahsa Mofidi, Abhilash Sahoo, Christopher J. Edelmaier.

**Project administration:** M. Gregory Forest, Ehssan Nazockdast, Sonya M. Hanson.

**Resources:** Sonya M. Hanson.

**Software:** S. Mahsa Mofidi.

**Supervision:** M. Gregory Forest, Ehssan Nazockdast, Sonya M. Hanson.

**Visualization:** S. Mahsa Mofidi.

**Writing – original draft:** S. Mahsa Mofidi.

**Writing – review & editing:** S. Mahsa Mofidi, Abhilash Sahoo, Stephen J. Klawa, Ronit Freeman, Amy Gladfelter, M. Gregory Forest, Ehssan Nazockdast, Sonya M. Hanson.

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
