## [Decision Letter · Decision Letter 0]

17 Nov 2025

PCOMPBIOL-D-25-01837

Study of Protein-Protein Interactions in Septin Assembly: Multiple amphipathic helix domains cooperate in binding to the lipid membrane

PLOS Computational Biology

Dear Dr. Hanson,

Thank you for submitting your manuscript to PLOS Computational Biology. After careful consideration, we feel that it has merit but does not fully meet PLOS Computational Biology's publication criteria as it currently stands. Therefore, we invite you to submit a revised version of the manuscript that addresses the points raised during the review process.

We look forward to receiving your revised manuscript.

Kind regards,

Muhammad Irfan

Academic Editor

PLOS Computational Biology

Arne Elofsson

Section Editor

PLOS Computational Biology

**Journal Requirements:**

At this stage, the following Authors/Authors require contributions: S. Mahsa Mofidi, Abhilash Sahoo, Christopher J. Edelmaier, Stephen J. Klawa, Ronit Freeman, Amy Gladfelter, M. Gregory Forest, Ehssan Nazockdast, and Sonya M. Hanson. Please ensure that the full contributions of each author are acknowledged in the "Add/Edit/Remove Authors" section of our submission form.

4) We notice that your supplementary Figures are included in the manuscript file. Please remove them and upload them with the file type 'Supporting Information'. Please ensure that each Supporting Information file has a legend listed in the manuscript after the references list.

State what role the funders took in the study. If the funders had no role in your study, please state: "The funders had no role in study design, data collection and analysis, decision to publish, or preparation of the manuscript.".

**Reviewers' comments:**

Reviewer's Responses to Questions

**Comments to the Authors:**

Reviewer #1: In their manuscript Mofidi et al. investigate the interactions of the amphipathic helix (AH) of Cdc12 in a lipid bilayer with molecular dynamics simulations. The experiments escalated logically from single peptide simulations to double peptide systems and other complexes. Taking the results together, the authors suggest a mechanism for how the Cdc12 of septin recruits other septins and thus starts the polymerization process. Most importantly, the authors highlight the charged residues involved in the process which could be readily validated via mutations in in vitro studies.

Overall the manuscript is written well and the figures have been presented elegantly. The following are my comments to improve the manuscript.

Major comments

1. I strongly recommend performing some coarse grained simulations with e.g. the Martini 3 force field to sample the peptide-peptide interactions. With the Martini force field the peptide tertiary structure has to be restrained so taking the average structure from lipid bound simulations could be a worthwhile strategy. The main message of this paper is the proposed mechanism for septin polymerization, and the four (out of eight that had binding) 400 ns simulations are very short to properly investigate the interaction. With so many degrees of freedom it is not surprising each simulation produced different and heterogeneous charge interactions. It would be highly valuable for in vitro validation to be able to point out a specific residue likely to participate in the interaction.

2. Both ends of the extended AH domain have been neutrally capped. However, its C-terminus matches the C-terminus of full length Cdc12 and thus unless Cdc12 is capped in vitro the C-terminus would be negatively charged, reducing the net charge from +3 to +2. It is unlikely this would significantly change the attained results, but should be mentioned as a limitation.

3. The use of a DOPC:PLPI 75:25 bilayer should be mentioned in the results or discussion and justified. Presumably it mimics the intracellular leaflet.

4. Charge interactions between the peptides and the different lipid species should be investigated. Beyond hydrophobic interactions perhaps specific residues drive the binding interaction, such as the positively charged C-terminal to negatively charged PLPI.

5. In figure 3A the distributions should be symmetrized to enable comparison. Presumably in one system a single peptide has migrated across the PBC to the negative side, reducing the distribution on the positive side.

6. The domain level contact maps of figure 4B should be shown for each replicate in the supplementary so that we can confirm each simulation reached a steady state. If the results are different between replicates the insufficient sampling should be mentioned as a limitation.

7. Since figure 4C is a representative example of how two peptides interact with each other, and it seems that the N-terminus of the floating peptide is unfolding, it should be mentioned as a limitation that the full length Cdc12 would influence the secondary structure and thus the interactions.

8. Final frame snapshots of the tandem peptide systems for each replicate should be shown in the supplementary. This would enable the reader to confirm the stability per system. Figure S8 shows these for one replicate, but four replicates were performed.

9. Consider publishing the full simulation trajectories in an open access repository like Zenodo.

Minor comments

10. Consider the visualization of the Z-COM boxplots. Currently the large amount of outliers merge all together taking focus away from the main message. Maybe omit the outliers or make them smaller. In Fig 3 B and C the Y-axis could be increased so that the distribution of early residues is visible. Additionally the grey line in the boxplots is not explained in the legend, presumably it is the phosphate peak.

12. To improve readability for a novice in the field the colour of lipids in the renders could be standardized. In figures 1 and 2 they are dark blue and cyan, in figure 3 red and grey, in figures 4 and 6 orange and white, and in figure 5 black and white.

13. In figures 4 and 6 the phospholipid tails have been rendered in some low resolution surface representation making them messy.

14. In figure 3A there is a typo in the legend for the green line.

15. In figure 5C there is a typo in the legend for the dark green line.

Reviewer #2: The authors report molecular dynamics simulation study of how single and paired extended

amphipathic helix (AH) domain of septins interact with lipid bilayer membranes and with

each other.

The novelty of the paper lies in providing molecular level insights into the

orientation of the AH domain on the membrane surface and the discovery that two AH domains

prefer to interact via an anti-parallel arrangment on the membrane surface, providing valuable

insights into septin higher-order assembly into filaments or other structures important

for various cellular functions.

Overall the paper is well organized and clearly written.

I like the paper as it is interesting to learn that AH domain interaction with each other

could be important in driving oligomer assembly on the membrane surface.

I would recommend publication after addressing the following issues.

Major issues/comments:

1. The authors claimed that a curved membrane may better accommodate multiple AH domains

(Abstract and Page 18). It is unclear how this claim is supported by the results, which

showed that the binding of multiple peptides induce lipid packing defects (increase in area/lipid and

decrease in lipid order parameter) but do not show stronger binding to regions of lipid packing defects.

Please provide some clarification on this.

2. The authors showed salt-bridge formation in contact map from Replica 7 for the system with bound + floating AH (Figure 4).

Is this or similar salt-bridges also exist from the contact maps for the other replicas (2, 4, 5 from Figure S6)?

Please show the contact maps for the other replicas for completeness.

3. The authors showed that when two peptides in parallel arrangement are embedded inside the membrane, in 75% of the cases the peptides remain bound to the membrane and interact with each other but in 25% of the cases one of the pair may become dissociated from the membrane (Page 15-16 and Figure 5B).

As the parallel arrangement was shown to be less stable than the anti-parallel one, please explain what is the rationale for using the parallel one?

Why not place the two peptides in the anti-parallel arrangement?

4. What is the effect of having more than two peptides on the inter-peptide interactions?

Specifically, how would an additional peptide bind to a peptide pair in anti-parallel fashion?

In anti-parallel or parallel fashion?

Some additional simulations to address this would be useful.

5: Are there examples of other AH domains that also form anti-parallel arrangement stabilized by

electrostatic interactions? Is anti-parallel AH domains a general motif that facilitates

higher order assemblies? Some discussion on this would be helpful.

Minor issues/comments:

1. In the first Results section the authors mentioned that simulations of isolated AH segment

(without flanking regions) do not show significant curvature (Page 6 Line 8). Please cite prior

simulation work (Edelmaier et al. as mentioned on lines 3-4 on Page 8?) or data.

Figures:

2. Figure 2D legend: "Helix" angle should have been "Bending" angle instead?

3. Figure 3A plot legend: peptide is misspelled as "peptied" for the green curve.

4. Figure 2E: The authors labeled the x-axis using Time Frame. It might be better to convert that

to simulation time or state how often each frame was saved (it seems to be a frame per 0.1 ns as

the total simulation time is 400 ns (see Table 1)).

5. Figure 3B and C: What are the black circles? The actual data points that contribute to the

box plot showing mean values? Please explain.

6. Figue 4C: It is somewhat difficult to identify the charged resides forming the salt-bridges

from the simulation snapshot. Perhaps the authors can include a more detailed zoomed-in view of

a few of the salt-bridges here?

7. Figure 4 legend for panel (D): The "n" in "number" should be capitalized.

8. Figure 6 legend for panel (B): The authors wrote "charged residues that are detrimental to forming salt-bridges".

Do they mean "instrumental" instead of "detrimental"?

Materials and methods:

9. It is not clear as to the choice of the lipid composition for the lipid bilayer.

Please explain why 75:25 DOPC:PLPI is used?

Although this seems to follow what some of the co-authors used in a prior publication

(Edelmaier et al. Biophysical Journal, 2025), some explanation/justification would still be

helpful to the readers as to understand why this particular composition is representative.

Reviewer #3: This manuscript uses all-atom molecular dynamics simulations to investigate how extended amphipathic helix (AH) domains of Cdc12 interact with lipid bilayers, how multiple AHs form peptide–peptide contacts (notably via salt bridges), and how these interactions may contribute to septin curvature sensing and cooperative assembly. The authors analyze single and double AH constructs (including tandems) in the presence of planar DOPC:PLPI bilayers and derive a mechanistic model for septin membrane recruitment and filament assembly. The work is potentially interesting and relevant to readers of PLOS Computational Biology, particularly those interested in membrane–protein interactions and cytoskeletal assembly. However, in its current form the manuscript over-interprets results obtained under relatively restricted simulation conditions, and several key analyses, controls, and clarifications are missing. I therefore recommend Major Revision before the manuscript can be considered for publication.

Major Concerns

1. The central narrative of the paper is that multiple AH domains “cooperate” in a way that is particularly suited to curved membranes, and the Discussion explicitly proposes a mechanism of septin polymerization on curved bilayers. However, all simulations appear to have been carried out on planar bilayers of a single composition (DOPC:PLPI 75:25). The conclusion that curved membranes “may better accommodate multiple AH domains” is therefore largely inferential. To support such a strong mechanistic statement, the authors should either:

(a) perform additional simulations on explicitly curved membranes (e.g., cylindrical or vesicular geometries, or at least buckled bilayers), and quantify how AH binding, peptide curvature, and lipid defects differ from the planar case; or

(b) substantially temper the claims throughout the manuscript, clearly labeling them as hypotheses rather than conclusions, and revising the title/abstract to avoid overstatement.

At present the text reads as if a curvature-sensing mechanism has been demonstrated, which is not fully supported by the presented data.

2.Each system is simulated for 400 ns with 4–8 replicas, but the manuscript provides relatively little information about convergence diagnostics, especially for key observables such as peptide curvature, salt-bridge occupancy, peptide–membrane residence times, and lipid order parameters.

• The authors should quantify and report time evolution and replica-to-replica variability for central metrics (e.g., radius of curvature, inter-peptide contact probabilities, lipid order/disorder near vs. far from the peptides).

• It would be helpful to show that the major structural states (e.g., anti-parallel vs. parallel arrangements) are robust to initial conditions and not dependent on a particular starting geometry or transient fluctuation.

• For the “dissociation” of one peptide in the two-bound system (25% of replicas), more rigorous analysis is needed to demonstrate that this is not simply a finite-time fluctuation within 400 ns.

Without such convergence and uncertainty analysis, it is difficult to judge how reliable the inferred “preferred” configurations and mechanistic inferences are.

3. The membrane patches are approximately 11 × 11 nm² with two AH peptides and relatively high local peptide density. This raises concerns that some of the observed effects—particularly the “upward shift” of peptide centers of mass and changes in APL and lipid order parameter maps—might be influenced by the small system size and periodic boundary conditions rather than generic membrane responses.

• Please justify the chosen membrane dimensions in light of the peptide size and number, and discuss possible box-size artifacts (e.g., peptide–peptide interactions through periodic images).

• Consider running at least one larger-patch control (or provide evidence from prior work) to demonstrate that the qualitative conclusions about lipid packing defects and peptide displacement are robust to box size.

4.The mechanistic model extrapolates from systems containing at most two AH peptides (or tandem constructs) to septin filaments and higher-order assemblies. Yet no simulations of longer arrays of AH domains or full septin oligomers are presented.

• The authors should more clearly delineate which conclusions are directly supported by their simulations (two peptides / tandem constructs) and which are speculative extrapolations to filament-scale behavior.

• If possible within reasonable computational cost, it would strengthen the manuscript to include at least one simulation containing a short linear array of three or more AHs to test whether the proposed anti-parallel salt-bridge motif can be propagated along a putative filament.

At minimum, the language in the Abstract, Author Summary, and Discussion should be moderated to avoid implying that filament assembly mechanisms have been directly observed.

5. Salt bridges between R and E residues (e.g., R4–E21) are presented as key drivers of AH–AH association and anti-parallel orientation. However, the analysis remains largely qualitative:

• Please provide quantitative occupancies (e.g., fraction of simulation time where each salt-bridge pair is formed) and associated lifetimes across all interacting replicas.

• Consider including radial distribution functions or contact probability matrices summarizing all charged residue pairs.

• It would be helpful to discuss how sensitive these salt-bridge patterns are to the chosen distance cutoffs and whether alternative definitions change the conclusions.

Given how central these interactions are for the proposed cooperative mechanism, a more rigorous statistical treatment is required.

6.The Data Availability statement indicates that full trajectories will be provided “upon reasonable request” due to storage and bandwidth limitations, whereas only processed data and representative frames are provided in the Supplementary Materials.

• Please clarify whether full or at least subsampled trajectories can be deposited in a public repository (e.g., Zenodo, OSF, or a domain-specific MD repository) with appropriate compression or frame thinning.

• If only a subset can be shared, specify clearly in the Data Availability statement what exactly is available, where, and how it suffices to reproduce all figures and key analyses.

7.The manuscript repeatedly references prior work from the authors on isolated AH domains and curvature sensing. It is sometimes difficult to distinguish which insights are genuinely new versus extensions or re-analyses of previously published simulations.

• Please more explicitly contrast the present work with previous studies, clearly stating what data and analyses are novel here (e.g., extended AH with flanking regions, explicit tandem constructs, two-peptide cooperative effects).

• Consider adding a short paragraph in the Introduction or Discussion that lists the main conceptual and technical advances beyond the prior study.

This clarification is important for assessing the incremental contribution of the current manuscript.

8.The APL and lipid order-parameter maps for single vs. double peptide systems are interesting, but the interpretation is again largely qualitative.

• Please provide quantitative statistics (mean ± SD over time and replicas) for APL and order parameters in defined regions (e.g., within a given radius of peptide vs. distal bulk membrane).

• Some of the proposed functional interpretations (e.g., “lateral pressure” leading to peptide displacement, or specific defect patterns being more compatible with curved membranes) would benefit from more cautious wording unless supported by additional analysis (e.g., lateral pressure profiles, free-energy calculations, or comparisons with curved systems).

Minor Comments

1.The Abstract and Author Summary are generally clear but somewhat dense, with several long sentences that mix methodological details and mechanistic conclusions. Consider tightening the text to distinguish more clearly between observed simulation results and proposed broader implications.

2.The manuscript uses several terms such as “extended AH,” “floating peptide,” and “bound peptide” that are not consistently defined at first use. A concise table or schematic summarizing all constructs (single AH, extended AH, NC–NC tandem, CN–NC tandem, extended CN–NC, etc.) and their sequences would improve readability.

3.There are several minor typographical errors (e.g., occasional misspellings and missing articles). A careful proofreading or copy-editing pass is recommended to improve polish.

4.Some figure panels (e.g., curvature heatmaps and multi-panel supplementary figures) have small fonts and color scales that are difficult to read, especially in print. Please ensure that axis labels, legends, and color bars are clearly legible at journal print size and that color choices are accessible (e.g., for color-blind readers).

5.The sequence logo and alignment used to argue for conservation of E and R residues involved in salt bridges are useful, but the Methods should provide more detail (number of sequences, selection criteria, species coverage) and the figure legend should explicitly mark the key residues discussed in the text.

6.The Methods specify specific ionic strength and lipid composition. It would be helpful to briefly justify these choices in the context of known yeast membrane composition and ionic strength, and to comment on how variations in composition or salt might alter the observed behavior.

7.The Bendix analysis uses a specific segment length to define local curvature. The text mentions a trade-off between resolution and smoothness, but it would be useful to show that the qualitative bending pattern is insensitive to moderate variations in this parameter (e.g., in supplementary information).

**Have the authors made all data and (if applicable) computational code underlying the findings in their manuscript fully available?**

The PLOS Data policy requires authors to make all data and code underlying the findings described in their manuscript fully available without restriction, with rare exception (please refer to the Data Availability Statement in the manuscript PDF file). The data and code should be provided as part of the manuscript or its supporting information, or deposited to a public repository. For example, in addition to summary statistics, the data points behind means, medians and variance measures should be available. If there are restrictions on publicly sharing data or code —e.g. participant privacy or use of data from a third party—those must be specified.requires authors to make all data and code underlying the findings described in their manuscript fully available without restriction, with rare exception (please refer to the Data Availability Statement in the manuscript PDF file). The data and code should be provided as part of the manuscript or its supporting information, or deposited to a public repository. For example, in addition to summary statistics, the data points behind means, medians and variance measures should be available. If there are restrictions on publicly sharing data or code —e.g. participant privacy or use of data from a third party—those must be specified.

Reviewer #1: **No:**Full simulation trajectories have not been made fully available.Full simulation trajectories have not been made fully available.

Reviewer #2: **No:**Only salt-bridge time series and helix bending angle data are available in the depository.Only salt-bridge time series and helix bending angle data are available in the depository.

Reviewer #3: **No:**The Data Availability statement indicates that full trajectories will be provided “upon reasonable request” due to storage and bandwidth limitations, whereas only processed data and representative frames are provided in the Supplementary Materials.The Data Availability statement indicates that full trajectories will be provided “upon reasonable request” due to storage and bandwidth limitations, whereas only processed data and representative frames are provided in the Supplementary Materials.

PLOS authors have the option to publish the peer review history of their article (what does this mean?). If published, this will include your full peer review and any attached files.). If published, this will include your full peer review and any attached files.

.

Reviewer #1: No

Reviewer #2: No

Reviewer #3: **Yes:**Yi ChenYi Chen

**Figure resubmission:**
---

## [Decision Letter · Decision Letter 1]

10 Apr 2026

Dear Dr. Hanson,

We are pleased to inform you that your manuscript 'Study of Protein-Protein Interactions in Septin Assembly: Multiple amphipathic helix domains cooperate in binding to the lipid membrane' has been provisionally accepted for publication in PLOS Computational Biology.

Best regards,

Muhammad Irfan

Academic Editor

PLOS Computational Biology

Arne Elofsson

Section Editor

PLOS Computational Biology

Reviewer's Responses to Questions

**Comments to the Authors:**

Reviewer #1: The authors have addressed all of my comments and I thank them for the effort.

Minor comment: I noticed that Figure S7B has replica 5 written twice.

Reviewer #2: Thank you for the revised manuscript. The authors have adequately addressed all of my concerns and the manuscript reads much better now.

**Have the authors made all data and (if applicable) computational code underlying the findings in their manuscript fully available?**

The PLOS Data policy requires authors to make all data and code underlying the findings described in their manuscript fully available without restriction, with rare exception (please refer to the Data Availability Statement in the manuscript PDF file). The data and code should be provided as part of the manuscript or its supporting information, or deposited to a public repository. For example, in addition to summary statistics, the data points behind means, medians and variance measures should be available. If there are restrictions on publicly sharing data or code —e.g. participant privacy or use of data from a third party—those must be specified.requires authors to make all data and code underlying the findings described in their manuscript fully available without restriction, with rare exception (please refer to the Data Availability Statement in the manuscript PDF file). The data and code should be provided as part of the manuscript or its supporting information, or deposited to a public repository. For example, in addition to summary statistics, the data points behind means, medians and variance measures should be available. If there are restrictions on publicly sharing data or code —e.g. participant privacy or use of data from a third party—those must be specified.

Reviewer #1: Yes

Reviewer #2: Yes

PLOS authors have the option to publish the peer review history of their article (what does this mean?). If published, this will include your full peer review and any attached files.). If published, this will include your full peer review and any attached files.

.

Reviewer #1: No

Reviewer #2: No

---

## [Editor Report · Acceptance letter]

PCOMPBIOL-D-25-01837R1

Study of Protein-Protein Interactions in Septin Assembly: Multiple amphipathic helix domains cooperate in binding to the lipid membrane

Dear Dr Hanson,

I am pleased to inform you that your manuscript has been formally accepted for publication in PLOS Computational Biology. Your manuscript is now with our production department and you will be notified of the publication date in due course.

With kind regards,

Judit Kozma
